# Best Response Regression

**Omer Ben-Porat**
Technion - Israel Institute of Technology
Haifa 32000 Israel
omerbp@campus.technion.ac.il

**Moshe Tennenholtz**
Technion - Israel Institute of Technology
Haifa 32000 Israel
moshet@ie.technion.ac.il

## Abstract

In a regression task, a predictor is given a set of instances, along with a real value for each point. Subsequently, she has to identify the value of a new instance as accurately as possible. In this work, we initiate the study of strategic predictions in machine learning. We consider a regression task tackled by two players, where the payoff of each player is the proportion of the points she predicts more accurately than the other player. We first revise the probably approximately correct learning framework to deal with the case of a duel between two predictors. We then devise an algorithm which finds a linear regression predictor that is a best response to any (not necessarily linear) regression algorithm. We show that it has linearithmic sample complexity, and polynomial time complexity when the dimension of the instances domain is fixed. We also test our approach in a high-dimensional setting, and show it significantly defeats classical regression algorithms in the prediction duel. Together, our work introduces a novel machine learning task that lends itself well to current competitive online settings, provides its theoretical foundations, and illustrates its applicability.

## 1 Introduction

Prediction is fundamental to machine learning and statistics. In a prediction task, an algorithm is given a sequence of examples composed of labeled instances, and its goal is to learn a general rule that maps instances to labels. When the labels take continuous values, the task is typically referred to as *regression*. The quality of a regression algorithm is measured by its success in predicting the value of an unlabeled instance. Literature on regression is mostly concerned with minimizing the discrepancy of the prediction, i.e. the difference between the true value and the predicted one.

Despite the tremendous amount of work on prediction and regression, online commerce presents new challenges. In this context, prediction is not carried out in isolation. New entrants can utilize knowledge of previous expert predictions and the corresponding true values, to maximize their probability of predicting better than that expert, treated as the new entrant's opponent. This fundamental task is the main challenge we tackle in this work.

We initiate the study of strategic predictions in machine learning. We present a regression learning setting that stems from a game-theoretic point of view, where the goal of the learner is to maximize the probability of being the most accurate among a set of predictors. Note that this approach may be in conflict with the traditional prediction goal.

Consider an online real estate expert, who frequently predicts the sale value of apartments. This expert, having been in the market for a while, has historical data on the values and characteristics of similar apartments. For simplicity, assume the expert uses simple linear regression to predict the value of an apartment as a function of its size. When a new apartment comes on the market, the expert uses her gathered historical data to predict the new apartment's value. When the apartment is sold, the true value (and the accuracy of the prediction) is revealed.

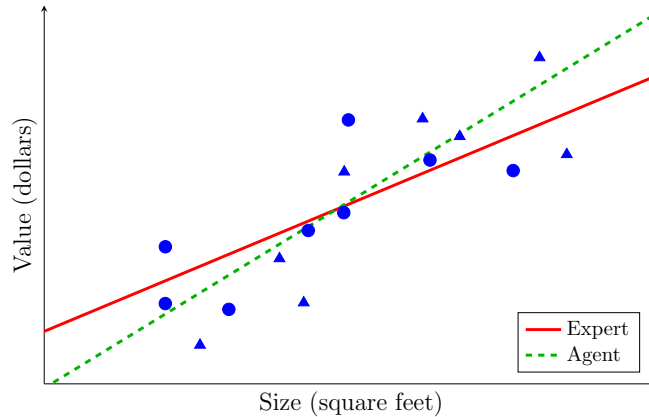

Figure 1: A case where minimizing the square error can be easily beaten. Each point is an instance-value pair, where the circles are historical points (i.e. their value has been revealed) and the triangles are new points, unseen by either the expert and the agent. The red (solid) line represents the linear least squares estimators, employed by the expert. After collecting a sufficient amount of historical data (circles) on apartments along with their true value and the value predicted by the expert, the agent comes up with the response represented by the green (dashed) line. For each of the unseen apartment sizes, both the expert and the agent declare their predictions of the apartment's value. Notice that the agent outperforms the expert in the majority of the historical points. In addition, the agent produces a more accurate prediction in the majority of the new (unseen) points.

At first glance this seems extremely effective, however it is also extremely fragile. An *agent* who enters the real estate business may come up with a linear predictor for which the probability (over all apartments and their values) of being more accurate is high, making it the preferable predictor. Figure 1 illustrates our approach. The expert uses linear least square estimators (LSE) to minimize the mean square error (MSE). The agent, after having collected "enough" historical data (circles) and having observed the predictions of the expert, produces a *strategy* (regression line). Both the expert and the agent predict the value of new apartments coming on the market (triangles). As illustrated, the prediction of the agent is the most accurate in the majority of new instances.

One criticism of this novel approach is that while maximizing the probability of being the most accurate, the agent may produce "embarrassing" predictions for some instances. Current prediction algorithms are designed to minimize some measure of overall loss, such as the MSE. Notice that in many, and perhaps even most, practical scenarios, being a better predictor on more instances is more important than avoiding such sporadic "embarrassing predictions". In particular, our approach fits any commerce and advertising setting where the agent offers predictions to users on the value of different goods or services, aiming at maximizing the number of users that will find her predictions more accurate than the one provided by the expert. For example, an agent, serving users searching for small apartments, would be happy to fail completely in predicting the value of very large sized apartments if this allowed predicting the value of smaller apartments better than an opponent.

Our novel perspective suggests several new fundamental problems:

1. Given a prediction algorithm ALG (e.g. LSE), what would be the best response to ALG, if we aim at maximizing the probability that the new algorithm would be more accurate than ALG?

2. In case ALG is unknown, but the agent has access to a labeled set of instances along with the prediction made by ALG for each instance, how many i.i.d. samples are needed in order to learn a best response to ALG over the whole population?

3. How poorly do classical regression algorithms preform against such a best response algorithm?

In this work, we focus on a two player scenario and analyze the best response of the agent against an opponent. We examine the agent's perspective, and introduce a rigorous treatment of Problems 1-3 above. We model the task of finding a best response as a supervised learning task, and show that it

fits the probably approximately correct (PAC) learning framework. Specifically, we show that when the strategy space of the agent is restricted, a best response over a large enough sample set is likely to be an approximate best response over the unknown distribution.

Our main result deals with an agent employing linear regression in $\mathbb{R}^n$ for any constant $n$. We present a polynomial time algorithm which computes a linear best response (i.e. from the set of all linear predictors) to *any* regression algorithm employed by the opponent. We also show a linearithmic bound in the number of training samples needed in order to successfully learn a best response. In addition, we show that in some cases our algorithm can be adapted to have an MSE score arbitrarily close to that of the given regression algorithm ALG. The theoretical analysis is complemented by an experimental study, which illustrates the effectiveness of our approach. In order to find a best linear response in high dimensional space, we provide a mixed integer linear programming (MILP) algorithm. The MILP algorithm is tested on the Boston housing dataset [5]. Indeed, we show that we can outperform classical regression algorithms in up to 70% of the points. Moreover, we outperform classical regression algorithms even in the case where they have full access to both training and test data, while we restrict our responder algorithm to the use of the training data only.

**Our contribution.** Our contributions are 3-fold. The main conceptual contribution of this paper is the explicit suggestion that a prediction task may have strategic aspects. We introduce the setting of best response regression, applicable to a huge variety of scenarios, and revise the PAC-learning framework to deal with such a duel framework. Then, we show an efficient algorithm dealing with finding a best-response linear regression in $\mathbb{R}^n$ for any constant $n$, against any regression algorithm. This best response algorithm maximizes the probability of beating the latter on new instances. Finally, we present an experimental study showing the applicability of our approach. Together, this work offers a new machine learning challenge, addresses some of its theoretical properties and algorithmic challenges, while also showing its applicability.

## 1.1 Related work

The intersection of learning theory with multi-agent systems is expanding with the rise of data science. In the field of mechanism design [8], [3, 7] considered prediction tasks with strategic aspects. In their model, the instances domain is to be labeled by one agent, and the dataset is constructed of points controlled by selfish users, who have their own view on how to label the instances domain. Hence, the users can misreport the points in order to sway decisions in their favor. A different line of work that is related to our model is the analysis of sample complexity in revenue maximizing auctions. In a recent work [2] the authors reconsider an auction setting where the auctioneer can sample from the valuation functions of the bidders, thereby relaxing the ubiquitous assumption of knowing the underlying distribution over bidders' valuations.

While the above papers consider mechanism design problems inspired by machine learning, our work considers a novel machine learning problem inspired by game theory.

In work on dueling algorithms [6], an optimization problem is analyzed from the perspective of competition, rather than from the point of view of a single optimizer. That work examines the dueling form of several optimization problems, e.g. the shortest path from the source vertex to the target vertex in a graph with random weights. While minimizing the expected length is a probable solution concept for a single optimizer, this is no longer the case in the defined duel. While [6] assumes a commonly-known distribution over a finite set of instances, we have no such assumption. Instead, we consider a sample set drawn from the underlying distribution with the aim of predicting a new instance better than the opponent.

Our formulation is also related to the Learning Using Privileged Information paradigm (see, e.g., [9, 14, 15]), in which the learner (agent) is supplied with additional information along with the label of each instance. In this paper, we assume the agent has access to predictions made by another algorithm (the opponent's), which can be treated as additional information.

## 2 Problem formulation

The environment is composed of instances and labels. In the motivating example given above, the instances are the characteristics of the apartments, and the labels are the values of these apartments.

A set of $N$ players offer predictive services, where a strategy of a player is a labeling function. For each instance-label pair $(x, y)$, the players see $x$, and subsequently each player $i$, predicts the value of the $y$. We call this label estimate $\hat{y}_i$. The player who wins a point $(x, y)$ is the one with the smallest discrepancy, i.e. $\min_i |\hat{y}_i - y|$. Under the strategy profile $(h_1, \ldots h_N)$, where each entry is the labeling function chosen by the corresponding player, the payoff of Player $i$ is $\Pr \left( \{ (x, y) : \text{Player } i \text{ wins } (x, y) \} \right)$.

A strategy of a player is called a *best response* if it maximizes the payoff of that player, when the strategies of all the other players are fixed. In this work, we analyze the best response of a player, and w.l.o.g. we assume she has only one opponent. The model is as follows:

1. We assume a distribution over the examples domain, which is the cross product of the instances domain $\mathcal{X} \subset \mathbb{R}^n$ and the labels domain $\mathcal{Y} \subset \mathbb{R}$.

2. The *agent* and the *opponent* both predict the label of each instance. The opponent uses a *strategy* $\bar{h}$, which is a conditional distribution over $\mathbb{R}$ given $x \in \mathcal{X}$.

3. The agent is unaware of the distribution over $\mathcal{X} \times \mathcal{Y}$ or the strategy of the opponent $\bar{h}$. Hence, we explicitly address the joint distribution $\mathcal{D}$ over $\mathcal{Z} = \mathcal{X} \times \mathcal{Y} \times \mathbb{R}$, where a triplet $(x, y, p)$ represents an instance $x$, its label $y$, and the discrepancy of the opponent's predicted value $p$, i.e. $p = |\bar{h}(x) - y|$. We stress that $\mathcal{D}$ is unknown to the agent.

4. The *payoff* of the agent under the strategy $h : \mathcal{X} \to \mathcal{Y}$ is given by
$$\pi_{\mathcal{D}}(h) = \mathbb{E}_{(x,y,p) \sim \mathcal{D}} \left( \mathbb{1}_{|h(x)-y)|<p} \right).$$

5. The agent has access to a sequence of examples $\mathcal{S}$, with which she wishes to maximize her payoff.

Note that a strategy which outputs $y_i$ for every instance $x_i$ in $\mathcal{S}$ may look promising, but will probably lead to overfitting, and low payoff for the agent. Since the agent wishes to generalize from $\mathcal{S}$ to $\mathcal{D}$, restricting the strategy set to $\mathcal{H} \subset \mathcal{Y}^{\mathcal{X}}$ seems justified. We define the goal of the agent:

6. The agent is willing to restrict herself to a strategy from $\mathcal{H} \subset \mathcal{Y}^{\mathcal{X}}$. Her goal: to find an algorithm which, given $\epsilon, \delta \in (0, 1)$ and a sequence of $m = m(\epsilon, \delta)$ examples $\mathcal{S}$ sampled i.i.d. from $\mathcal{D}$, outputs a strategy $h^*$ such that with probability at least $1 - \delta$ (over the choices of $\mathcal{S}$) it holds that
$$\pi_{\mathcal{D}}(h^*) \geq \sup_{h \in \mathcal{H}} \pi_{\mathcal{D}}(h) - \epsilon.$$

Indeed, the access to a sequence of examples seems realistic, and the size of $\mathcal{S}$ depends on the amount of resources at the agent's disposal. The size of $\mathcal{S}$ also affects the selection of $\mathcal{H}$: if the agent can gather "many" examples, she might be able to learn a "good" strategy from a more complex strategy space.

We say that $h \in \mathcal{H}$ is an *approximate best response* with factor $\epsilon$ if for all $h' \in \mathcal{H}$ it holds that $\pi_{\mathcal{D}}(h') - \pi_{\mathcal{D}}(h) \leq \epsilon$. Note that the goal of the agent can be interpreted as finding an approximate best response with high probability. The *empirical payoff* of the agent is defined by
$$\pi_{\mathcal{S}}(h) = \frac{1}{m} \cdot \left| \{ i : \mathbb{1}_{|h(x_i)-y_i)|<p_i} \} \right|,$$

and a strategy $h \in \arg\max_{h' \in \mathcal{H}} \pi_{\mathcal{S}}(h')$ is called an *empirical best response* (w.r.t $\mathcal{S}$). Next, we adopt the PAC framework [12] to define under which strategy spaces an empirical best response is likely to be an approximate best response.

## 2.1 Approximate best response with PAC learnability

The field of statistical learning addresses the problem of finding a predictive function based on data. We briefly define some key concepts in learning theory, that will be used later. For a more gentle introduction the reader is referred to [11].

Let $\mathcal{G}$ be a class of functions from $\mathcal{Z}$ to $\{0, 1\}$ and let $\mathcal{S} = \{z_1, \ldots, z_m\} \subset \mathcal{Z}$. The *restriction* of $\mathcal{G}$ to $\mathcal{S}$, denoted $\mathcal{G}(\mathcal{S})$, is defined by $\mathcal{G}(\mathcal{S}) = \{ (g(z_1), g(z_2), \ldots, g(z_m)) : g \in \mathcal{G} \}$. Namely, $\mathcal{G}(\mathcal{S})$ contains all the binary vectors induced by the functions in $\mathcal{G}$ on the items of $\mathcal{S}$. We say that $\mathcal{G}$ *shatters* $\mathcal{S}$ if $\mathcal{G}(\mathcal{S})$ contains all binary vectors of size $m$, i.e. $|\mathcal{G}(\mathcal{S})| = 2^m$.

**Definition 1** (VC dimension,[13]). *The VC dimension of a class $\mathcal{G}$, denoted* $\text{VCdim}(\mathcal{G})$*, is the maximal size of a set $\mathcal{S} \subset \mathcal{Z}$ that can be shattered by $\mathcal{G}$.*

**Definition 2** (PAC learnability,[12]). *A hypothesis class $\mathcal{H}$ is PAC-learnable with respect to a domain set $\mathcal{Z}$ and a loss function $l : \mathcal{H} \times \mathcal{Z} \rightarrow \mathbb{R}_+$ , if there exists a function $\tau_{\mathcal{H}} : (0,1)^2 \rightarrow \mathbb{N}$ and a learning algorithm ALG such that for every $\epsilon, \delta \in (0,1)$ and for every distribution $\mathcal{D}$ over $\mathcal{Z}$, when running ALG on $m \geq \tau_{\mathcal{H}}(\epsilon, \delta)$ i.i.d. examples generated by $\mathcal{D}$, it returns a hypothesis $h \in \mathcal{H}$ such that with probability of at least $1 - \delta$ it holds that*

$$L_{\mathcal{D}}(h) \leq \inf_{h' \in \mathcal{H}} L_{\mathcal{D}}(h') + \epsilon, \tag{1}$$

*where $L_{\mathcal{D}}(h) = E_{z \sim \mathcal{Z}} l(h, z)$.*

Let $\mathcal{H}$ be a class of functions from $\mathcal{X}$ to $\mathcal{Y}$, and let $\mathcal{Z} = \mathcal{X} \times \mathcal{Y} \times \mathbb{R}$, as defined earlier in this section. Typically in a regression task, the hypothesis class is restricted in order to decrease the distance between the predicted labels and the true label. In the aforementioned model, however, the agent may want to deliberately harm her accuracy on some subset of the instances domain. She will do this as long as it increases the number of instances having a better prediction, thereby improving her payoff.

Since $h \in \mathcal{H}$ can either win a point $(x, y, p)$ or lose it, the model resembles a binary classification task, where the "label" of $(x, y, p)$ is the identity of the winner. That is, a triplet $(x, y, p)$ would be labeled 1 if the agent produced a better prediction than the opponent, and zero otherwise. However, notice that the agent's strategy is involved in the labeling. This is, of course, not the case of binary classification. Our approach is to introduce a corresponding binary classification problem, and by leveraging former results obtained on binary classification, deduce sufficient learnability conditions for our model. The complete reduction is described in detail in the appendix.

Adjusting to the loss function framework, define:

$$\forall z = (x, y, p) \in \mathcal{Z} : l(h, z) = \begin{cases} 1 & |h(x) - y| \geq p \\ 0 & |h(x) - y| < p \end{cases}.$$

Observe that $l(h, z) = 0$ whenever the agent wins a point and $l(h, z) = 1$ otherwise. If we set $L_{\mathcal{D}}(h) = \mathbb{E}_{z \sim \mathcal{D}} l(h, z)$, Equation (1) can be reformulated as $\pi_{\mathcal{D}}(h) \geq \sup_{h' \in \mathcal{H}} \pi_{\mathcal{D}}(h') - \epsilon$. Our goal is to find sufficient conditions for $\mathcal{H}$ to be PAC-learnable w.r.t $\mathcal{Z}$ and $l$.

Given $\mathcal{H}$, let $\mathcal{G}_{\mathcal{H}} = \{g_h : h \in \mathcal{H}\}$ such that

$$\forall h \in \mathcal{H}, \forall z \in \mathcal{Z} : g_h(z) = 1 - l(h, z) = \begin{cases} 1 & |h(x) - y| < p \\ 0 & |h(x) - y| \geq p \end{cases}.$$

Note that $\mathcal{G}_{\mathcal{H}}$ is a class of functions from $\mathcal{Z}$ to $\{0, 1\}$. Sufficient learnability conditions can now be stated.

**Lemma 1.** *Let $\mathcal{H}$ be a class of functions from $\mathcal{X}$ to $\mathcal{Y}$ with $\text{VCdim}(\mathcal{G}_{\mathcal{H}}) = d < \infty$. Then there is a constant $C$, such that for every $\epsilon, \delta \in (0,1)$ and every distribution $\mathcal{D}$ over $\mathcal{Z} = \mathcal{X} \times \mathcal{Y} \times \mathbb{R}$, if we sample a sequence of examples $\mathcal{S}$ of size $m \geq C \cdot \frac{d + \log \frac{1}{\delta}}{\epsilon^2}$ i.i.d. from $\mathcal{D}$ and pick an empirical best response $h \in \mathcal{H}$ w.r.t. $\mathcal{S}$, then with probability of at least $1 - \delta$ it holds that*

$$\pi_{\mathcal{D}}(h) \geq \sup_{h' \in \mathcal{H}} \pi_{\mathcal{D}}(h') - \epsilon.$$

## 3   Best linear response

We assume throughout this section that the agent uses a linear response. In what follows, we first show that $\mathcal{H}$ is PAC-learnable with respect to $\mathcal{Z}$ and the payoff function. Afterwards, we devise an empirical best response algorithm with respect to a sequence of examples. Hence, according to the previous section, this empirical payoff maximization algorithm outputs, with high probability, an approximate best response with respect to $\mathcal{D}$. The proofs of all theorems and the supporting lemmas are in the appendix.

For ease of presentation, we re-denote the dimension of the instances domain to be $n - 1$, i.e. $\mathcal{X} \subset \mathbb{R}^{n-1}$. Every $\boldsymbol{h} \in \mathbb{R}^n$ defines a linear predictor of a point $\boldsymbol{x} \in \mathbb{R}^{n-1}$ via dot product, namely

$h \cdot (x_i, 1)$. Thus, $\mathbb{R}^n$ is referred to as the strategy space $\mathcal{H}$, where axis $i$ represents the $i$'th entry in $h$, $1 \le i \le n+1$. We study the case where $n$ is fixed, although the complementary case is discussed in the end of the section.

Recall that the empirical payoff of the agent w.r.t to a sequence of examples $\mathcal{S} = (x_i, y_i, p_i)_{i=1}^m$ is defined as $\pi_{\mathcal{S}}(h) = \frac{1}{m} \sum_{i=1}^m \mathbb{1}_{|h \cdot (x_i, 1) - y_i| < p_i}$, and the best response w.r.t. to $\mathcal{S}$ is $\arg\max_{h \in \mathcal{H}} \pi_{\mathcal{S}}(h)$. Observe that there is a mapping $\mathcal{M}_{\mathcal{S}}^{\mathcal{H}} : \mathcal{H} \to \{0,1\}^m$ from any $h \in \mathcal{H}$ to a vector $v \in \{0,1\}^m$ such that entry $i$ in $v$ equals one if $h$ gains the $i$'th point, and zero otherwise. Put differently, $\mathcal{M}_{\mathcal{S}}^{\mathcal{H}}(h) = v = (v_1, \ldots v_m)$ such that:

$$\forall i \in [m] : v_i = 1 \Leftrightarrow |h \cdot (x_i, 1) - y_i| < p_i.$$

Hence, the target set of $\mathcal{M}_{\mathcal{S}}^{\mathcal{H}}$ is $\mathcal{G}_{\mathcal{H}}(\mathcal{S})$, which is the restriction of $\mathcal{G}_{\mathcal{H}}$ to $\mathcal{S}$. The size of $\mathcal{G}_{\mathcal{H}}(\mathcal{S})$ is essentially the effective size of $\mathcal{H}$, since any two strategies which are mapped to the same vector will gain the same points, and thus are equivalent. The following theorem puts a bound on the size of $\mathcal{G}_{\mathcal{H}}(\mathcal{S})$.

**Theorem 1.** *Let $\mathcal{H}$ be the hypothesis class of all linear functions in $\mathbb{R}^{n-1}$. For any sequence of examples $\mathcal{S}$ of size $m$, $\mathcal{G}_{\mathcal{H}}(\mathcal{S})$ is polynomial in $m$. Specifically, $|\mathcal{G}_{\mathcal{H}}(\mathcal{S})| \le \sum_{i=0}^n 2^i \binom{m}{i}$.*

The VC-dimension of $\mathcal{G}_{\mathcal{H}}$ can be bounded using the Sauer - Shelah lemma [10]:

**Lemma 2.** *It holds that $\text{VCdim}(\mathcal{G}_{\mathcal{H}}) \le \max\{\lfloor 2n \cdot log(n) \rfloor, 20\}$.*

We now devise an empirical payoff maximizing algorithm. Our approach is to first explicitly characterize the vectors in $\mathcal{G}_{\mathcal{H}}(\mathcal{S})$, and afterwards to pick a strategy from

$$\left\{ h : \left\| \mathcal{M}_{\mathcal{S}}^{\mathcal{H}}(h) \right\|_1 = \max_{v \in \mathcal{G}_{\mathcal{H}}(\mathcal{S})} \|v\|_1 \right\}.$$

For each vector $v$, one can formulate a linear program which outputs a strategy in $\{h : \mathcal{M}_{\mathcal{S}}^{\mathcal{H}}(h) = v\}$ in case this set is not empty, or outputs none in case it is. Naively, $2^m$ such feasibility problems can be solved, although this is very inefficient. Instead, we will recursively construct the set of feasible vectors. The Partial Vector Feasibility problem aids in recursively partitioning the hypothesis space. Note that it is solvable in time $poly(n, m)$ using Linear Programming.

---

**Problem:** PARTIAL VECTOR FEASIBILITY (PVF)

    **Input**: a sequence of examples $\mathcal{S} = (x_i, y_i, p_i)_{i=1}^m$, and a vector $v \in \{1, 0, a, b\}^m$
    **Output**: a point $h \in \mathbb{R}^n$ satisfying

        1. If $v_i = 1$ then $|h \cdot (x_i, 1) - y_i| < p_i$.

        2. If $v_i = a$ then $h \cdot (x_i, 1) - y_i > p_i$.

        3. If $v_i = b$ then $h \cdot (x_i, 1) - y_i < -p_i$.

    if such exists, and $\phi$ otherwise.

---

The following algorithm partitions $\mathbb{R}^n$ according to $\mathcal{G}_{\mathcal{H}}(\mathcal{S})$, where in each iteration it "discovers" one more point in the sequence $\mathcal{S}$.

---

**Algorithm:** EMPIRICAL PAYOFF MAXIMIZATION (EPM)

    **Input**: $\mathcal{S} = (x_i, y_i, p_i)_{i=1}^m$
    **Output**: Empirical payoff maximizer w.r.t. $\mathcal{S}$
**1**   $v \leftarrow \{0\}^m$                           // $v = (v_1, v_2, \ldots, v_m)$
**2**   $\mathcal{R}_0 \leftarrow \{v\}$
**3**   **for** $i = 1$ *to* $m$ **do**
**4**      $\mathcal{R}_i \leftarrow \emptyset$
**5**      **for** $v \in \mathcal{R}_{i-1}$ **do**
**6**          **for** $\alpha \in \{1, a, b\}$ **do**
**7**              **if** $\text{PVF}(\mathcal{S}, (v_{-i}, \alpha)) \ne \phi$ **then**
**8**                 add $(v_{-i}, \alpha)$ to $\mathcal{R}_i$    // $(v_{-i}, \alpha) = (v_1, \ldots v_{i-1}, \alpha, v_{i+1}, \ldots, v_m)$
**9**   **return** $v^* \in \arg\max_{v \in \mathcal{R}_m} \|v\|_1$

---

**Theorem 2.** *When running* EPM *on a sequence of examples $\mathcal{S}$, it finds an empirical best response in $poly(|\mathcal{S}|)$ time.*

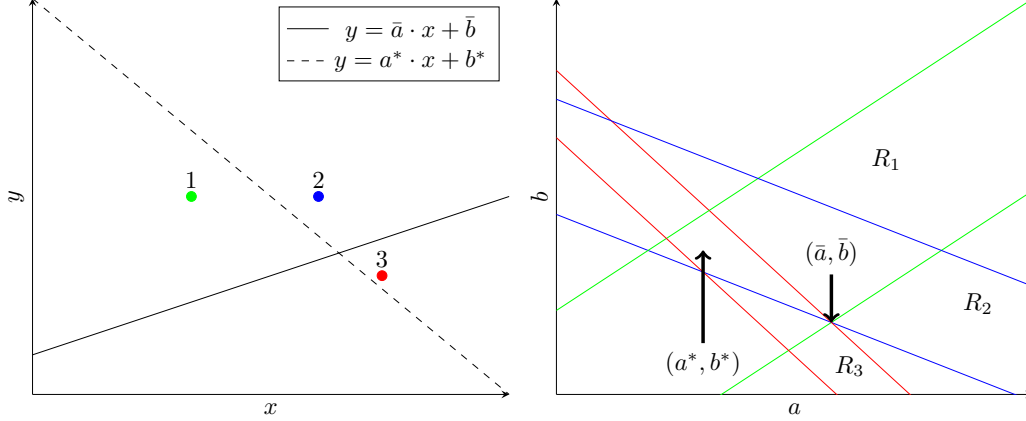

Figure 2: An example of simple linear regression with linear strategies. On the left we have a sample sequence of size 3, along with the strategy $\bar{\boldsymbol{h}} = (\bar{a}, \bar{b})$ of the opponent (the solid line) and a best response strategy of the agent (the dashed line). On the right the hypothesis space is presented, where each pair $(a, b)$ represents a possible strategy, and each bounded set $R_i$ is defined by $R_i = \{(a, b) \in \mathbb{R}^2 : |a \cdot x_i + b - y_i| < p_i\}$, i.e. the set of hypotheses which give $x_i$ better prediction than $\bar{h}$. Notice that $(\bar{a}, \bar{b})$ relies on the boundaries of all $R_i, 1 \leq i \leq 3$. In addition, since $(a^*, b^*)$ is inside $R_1 \cap R_2 \cap R_3$, the strategy $\boldsymbol{h}^* = (a^*, b^*)$, i.e. the line $y = a^* \cdot x + b^*$, predicts all the points better than the opponent. Observe that by taking any convex combination of $\boldsymbol{h}^*, \bar{\boldsymbol{h}}$, the agent not only perserves her empirical payoff but also improves her MSE score.

When we combine Theorem 2 with Lemmas 2 and 1, we get:

**Corollary 1.** *Given $\epsilon, \delta \in (0, 1)$, if we run EPM on $m \geq \frac{C}{\epsilon^2} \cdot \left( \max\{\lfloor 2n \cdot log(n) \rfloor, 20\} + \log \frac{1}{\delta} \right)$ examples sampled i.i.d. from $\mathcal{D}$ (for a constant $C$), then it outputs $h^*$ such that with probability at least $1 - \delta$ satisfies*

$$\pi_{\mathcal{D}}(h^*) \geq \sup_{h' \in \mathcal{H}} \pi_{\mathcal{D}}(h') - \epsilon.$$

A desirable achievement would be if the best response prediction algorithm would also keep the loss small in the original (e.g. MSE) measure. We now show that in some cases the agent can, by slightly modifying the output of EPM, find a strategy that is not only an approximate best response, but is also robust with respect to additive functions of discrepancies. See Figure 2 for illustration.

**Lemma 3.** *Assume the opponent uses a linear predictor $\bar{h}$, and denote by $\boldsymbol{h}^*$ the strategy output by EPM. Then, $\boldsymbol{h}^*$ can be efficiently modified to a strategy which is not only an empirical best response, but also performs arbitrarily close to $\bar{h}$ w.r.t. to any additive function of the discrepancies.*

Finaly, we discuss the case where the dimension of the instances domain is a part of the input. It is known that learning the best halfspace is NP-hard in binary classification (w.r.t. to a given sequence of points), when the dimension of the data is not fixed (see e.g. [1]). We show that the empirical best (linear) response problem is of the same flavor.

**Lemma 4.** *In case $\mathcal{H}$ is the set of linear functions in $\mathbb{R}^{n-1}$ and $n$ is not fixed, the empirical best response problem is NP-hard.*

## 4 Experimental results

We note that when $n$ is large, the proposed method for finding an empirical best response may not be suitable. Nevertheless, if the agent is interested in finding a "good" response to her opponents, she should come up with *something*. With slight modifications, the linear best response problem can be formulated as a mixed integer linear program (MILP).[1] Hence, the agent can exploit sophisticated solvers and use clever heuristics. Further, one implication of Lemma 1 is that the true payoffs

Table 1: Experiments on Boston Housing dataset

| The opponent's strategy | Scenario | Train payoff | Test payoff |
|---|---|---|---|
| Least square errors (LSE) | TRAIN | 0.699 | 0.641 |
| | ALL | 0.711 | 0.645 |
| Least absolute errors (LAE) | TRAIN | 0.621 | 0.570 |
| | ALL | 0.625 | 0.528 |

Results obtained on the Boston Housing dataset. Each cell in the table represents the average payoff of the agent over 1000 simulations (splits into 80% train and 20% test). The "train payoff" is the proportion of points in the training set on which the agent is more accurate, and the "test payoff" payoff is the equivalent proportion with respect to the test (unseen) data.

uniformly converge, and hence any empirical payoff obtained by the MILP is close to its real payoff with high probability.

In this section, we show the extent to which classical linear regression algorithms can be beaten using the Boston housing dataset [5], a built-in dataset in the leading data science packages (e.g. scikit-learn in Python and MASS in R). The Boston housing dataset contains 506 instances, where each instance has 13 continuous attributes and one binary attribute. The label is the median value of owner-occupied homes, and among the attributes are the per capita crime rate, the average number of rooms per dwelling, the pupil-teacher ratio by town and more. The R-squared measure for minimizing the square error in the Boston housing dataset is 0.74, indicating that the use of linear regression is reasonable.

As possible strategies of the opponent, we analyzed the linear least squares estimators (LSE) and linear least absolute estimators (LAE). The dataset was split into training (80%) and test (20%) sets, and two scenarios were considered:

Scenario TRAIN - the opponent's model is learned from the training set only.

Scenario ALL - the opponent's model is learned from both the training and the test sets.

In both scenarios the agent had access to the training set only, along with the opponent's discrepancy for each point in the training set. Obviously, achieving payoff of more than 0.5 (that is, more than 50% of the points) in the ALL scenario is a real challenge, since the opponent has seen the test set in her learning process. We ran 1000 simulations, where each simulation is a random split of the dataset. We employed the MILP formulation, and used Gurobi software [4] in order to find a response, where the running time of the solver was limited to one minute.[2]

Our findings are reported in Table 1. Notice that against both opponent strategies, and even in case where the opponent had seen the test set, the agent still gets more than 50% of the points. In both scenarios, LAE guarantees the opponent more than LSE. This is because absolute error is less sensitive to large deviations. We also noticed that when the opponent learns from the whole dataset, the empirical payoff of the agent is greater. Indeed, the latter is reasonable as in the ALL scenario the agent's strategy fits the training set while the opponent strategy does not.

Beyond the main analysis, we examined the success (or lack thereof) of the agent with respect to the additive loss function optimized by the opponent (corresponding to the MSE for LSE, and the MAE (mean absolute error) for LAE), hereby referred to as the "classical loss". Recall that Lemma 3 guarantees that the agent's classical loss can be arbitrarily close to that of the opponent when she plays a best response; however, the response we consider in this section (using the MILP) does not necessarily converge to a best response. Therefore, we find it interesting to consider the classical loss as well, thereby presenting the complementary view.

We report in Table 2 the average ratio between the agent's classical loss and that of the opponent under the TRAIN scenario with respect to the training and test sets. Notice that the agent suffers from less than a 0.7% increase with respect to the classical loss optimized by the opponent. In particular,

Table 2: Ratio of the classical loss

| | The opponent's strategy | |
|---|---|---|
| | LSE | LAE |
| Training set | 1.007 | 1.005 |
| Test set | 0.999 | 1.002 |

Ratio of the agent's loss and the opponent's loss, where the loss function corresponds to the original optimization function of the opponent, under scenario TRAIN. For example, the upper leftmost cell represents the agent's MSE divided by the opponents MSE on the training set, where the opponent uses LSE. Similarly, the lower rightmost cell represents the agent's MAE (mean absolute error) divided by the opponents MAE on the test data, when the opponent uses LAE.

the MSE of the agent (when she responds to LSE) on the test set is less than that of the opponent. The same phenomenon, albeit on a smaller scale, occurs against LAE: the training set ratio is greater than the test set ratio.

To conclude, the agent is not only able to obtain the majority of the points (and in some cases, up to 70%), but also to keep the classical loss optimized by her opponent within less than 0.2% from the optimum on the test set.

## 5   Discussion

This work introduces a game theoretic view of a machine learning task. After finding sufficient conditions for learning to occur, we analyzed the induced learning problem, when the agent is restricted to a linear response. We showed that a best response with respect to a sequence of examples can be computed in polynomial time in the number of examples, as long as the instance domain has a constant dimension. Further, we showed an algorithm that for any $\epsilon, \delta$ computes an $\epsilon$-best response with a probability of at least $1 - \delta$, when it is given a sequence of poly $\left( \frac{1}{\epsilon^2} \left( n \log n + \log \frac{1}{\delta} \right) \right)$ examples drawn i.i.d.

As the reader may notice, our analysis holds as long as the hypothesis is linear in its parameters, and therefore is much more general than linear regression. Interestingly, this is a novel type of optimization problem and so rich hypothesis, which are somewhat unnatural in the traditional task of regression, might be successfully employed in the proposed setting.

From an empirical standpoint, the gap between the empirical payoff and the true payoff calls for applying regularization methods for the best response problem and encourages further algorithmic research. Exploring whether or not a response in the form of hyperplanes can be effective against a more complex strategy employed by the opponent will be intriguing. For instance, showing that a deep learner is beatable in this setting will be remarkable.

The main direction to follow is the analysis of the competitive environment introduced in the beginning of Section 2 as a simultaneous game: is there an equilibrium strategy? Namely, is there a linear predictor which, when used by both the agent and the opponent, is a best response to one another?

**Acknowledgments**

We thank Gili Baumer and Argyris Deligkas for helpful discussions, and anonymous reviewers for their useful suggestions. This project has received funding from the European Research Council (ERC) under the European Union's Horizon 2020 research and innovation programme (grant agreement n° 740435).

## Footnotes

[1]See the appendix for the mixed integer linear programming formulation.

[2]Code for reproducing the experiments is available at `https://github.com/omerbp/Best-Response-Regression`

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
