[Supplementary Material · nips2017-brr-cr-supp.pdf]

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

**Proof of Lemma 1**  For convenience, we restate the definition of $\mathcal{G}_{\mathcal{H}}$: Let $\mathcal{G}_{\mathcal{H}} = \{g_h : h \in \mathcal{H}\}$ such that

$$\forall h \in \mathcal{H}, \forall z \in \mathcal{Z} : g_h(z) = 1 - l(h, z) = \begin{cases} 1 & |h(x) - y| < p \\ 0 & |h(x) - y| \geq p \end{cases}.$$

In the following analysis, we use the tilde notation to denote the objects of the binary classification problem.

For $\tilde{\mathcal{Z}} = \mathcal{Z} \times \{0,1\}$, a distribution $\tilde{\mathcal{D}}$ over $\tilde{\mathcal{Z}}$ and a sequence of examples $\tilde{\mathcal{S}} = (z_i, \tilde{y}_i)_{i=1}^m$ of size $m$ drawn i.i.d. from $\tilde{\mathcal{D}}$, define:

$$\tilde{L}_{\tilde{\mathcal{S}}}(g_h) = \frac{1}{m} \sum_{i=1}^m \mathbb{1}_{g_h(z_i) \neq \tilde{y}_i} \quad , \quad \tilde{L}_{\tilde{\mathcal{D}}}(g_h) = \mathop{\mathbb{E}}_{(z,\tilde{y}) \sim \tilde{\mathcal{D}}} \left( \mathbb{1}_{g_h(z) \neq \tilde{y}} \right).$$

$\tilde{L}_{\tilde{\mathcal{S}}}$ is called the *empirical risk*, and an algorithm that minimizes $\tilde{L}_{\tilde{\mathcal{S}}}$ for any $\tilde{\mathcal{S}}$ is called empirical risk minimizer (ERM). A fundamental result in the field of learning theory is the following [2]:

**Theorem 3.** *Let $\mathcal{G}$ be a hypothesis class of functions from $\mathcal{Z}$ to $\{0,1\}$, let the loss function be the 0-1 loss, and assume that $\mathrm{VCdim}(\mathcal{G}) = d < \infty$. There is a constant $C$, such that for every distribution $\tilde{\mathcal{D}}$ and for every $\epsilon, \delta \in (0,1)$, when taking an ERM hypothesis $g_h \in \mathcal{G}$ over $m \geq C \cdot \frac{d + \log \frac{1}{\delta}}{\epsilon^2}$ examples sampled i.i.d. from $\tilde{\mathcal{D}}$, with probability of at least $1 - \delta$ it holds that*

$$\tilde{L}_{\tilde{\mathcal{D}}}(g_h) \leq \inf_{g_h' \in \mathcal{G}} \tilde{L}_{\tilde{\mathcal{D}}}(g_h') + \epsilon. \tag{2}$$

Equation (2) holds for every distribution $\tilde{\mathcal{D}}$ over $\mathcal{Z} \times \{0,1\}$, and in particular for $\tilde{\mathcal{D}}_0$, which is defined by

$$\forall B \subseteq \mathcal{Z} : \tilde{\mathcal{D}}_0(B, \tilde{y}) = \begin{cases} \mathcal{D}(B) & \tilde{y} = 1 \\ 0 & \tilde{y} = 0 \end{cases}.$$

Note that classification is trivial under $\tilde{\mathcal{D}}_0$, since every instance $z \in \mathcal{Z}$ gets the label 1. Nevertheless, since the class of hypothesis considered is $\mathcal{G}_{\mathcal{H}}$, finding an ERM is a greater challenge. Observe that under $\tilde{\mathcal{D}}_0$:

$$\tilde{L}_{\tilde{\mathcal{D}}_0}(g_h) = \mathop{\mathbb{E}}_{(z,\tilde{y}) \sim \tilde{\mathcal{D}}_0} \left( \mathbb{1}_{g_h(z) \neq \tilde{y}} \right) = \mathop{\mathbb{E}}_{z \sim \mathcal{D}} \left( \mathbb{1}_{g_h(z) \neq 1} \right) = \mathop{\mathbb{E}}_{z \sim \mathcal{D}} \left( l(h, z) \right) = L_{\mathcal{D}}(h).$$

Further, if $\tilde{\mathcal{S}} = (\tilde{z}_i)_{i=1}^m$ is a sequence of examples sampled from $\tilde{\mathcal{D}}_0$, then $\tilde{z}_i = z_i \oplus 1$, hence if we denote $\mathcal{S} = (z_i)_{i=1}^m$ we have:

**Proposition 1.**
$$h^* \in \arg\min_{h \in \mathcal{H}} L_{\mathcal{S}}(h) \iff g_{h^*} \in \arg\min_{g_h \in \mathcal{G}_{\mathcal{H}}} \tilde{L}_{\tilde{\mathcal{S}}}(g_h).$$

*Proof.*
$$h^* \in \arg\min_{h \in \mathcal{H}} L_{\mathcal{S}}(h) \Leftrightarrow$$
$$\forall h \in \mathcal{H} : L_{\mathcal{S}}(h) \geq L_{\mathcal{S}}(h^*) \Leftrightarrow$$
$$\forall g_h \in \mathcal{G}_{\mathcal{H}} : \tilde{L}_{\tilde{\mathcal{S}}}(g_h) \geq \tilde{L}_{\tilde{\mathcal{S}}}(g_{h^*}) \Leftrightarrow$$
$$g_{h^*} \in \arg\min_{g_h \in \mathcal{G}_{\mathcal{H}}} \tilde{L}_{\tilde{\mathcal{S}}}(g_h).$$

$\square$

Thus, by choosing ERM for $L_{\mathcal{S}}$ we are essentially minimizing $\tilde{L}_{\tilde{\mathcal{S}}}$. This concludes the proof of the lemma.

# 6   Omitted proofs from Section 3

The tools we use for the proofs involve arrangement of hyperplanes. Although self-contained, we refer the curious reader to [14, Chapter 3].

Every $\boldsymbol{h} \in \mathbb{R}^n$ defines a linear predictor of a point $\boldsymbol{x} \in \mathbb{R}^{n-1}$ via dot product, i.e. $\boldsymbol{h} \cdot (\boldsymbol{x}_i, 1) = y_i$, and we refer to this space as the *parameter space*, where axis $i$ represents the $i$'th entry in $\boldsymbol{h}$, $1 \leq i \leq n$. Fix a sequence of examples $\mathcal{S} = (x_i, y_i, p_i)_{i=1}^m$.

The forthcoming analysis treats every triplet $(x_i, y_i, p_i)$ separately, by asking which strategies of the agent, i.e. elements in $\mathcal{H} = \mathbb{R}^n$, will entail true value for the indicator $\mathbb{1}_{|\boldsymbol{h} \cdot (\boldsymbol{x}_i, 1) - y_i| < p_i}$.

Denote:
$$R_i = \{\boldsymbol{h} \in \mathbb{R}^n : |\boldsymbol{h} \cdot (\boldsymbol{x}_i, 1) - y_i| < p_i\}.$$

The empirical payoff of the agent under the strategy $\boldsymbol{h}$ w.r.t. to $\mathcal{S}$ can thus be interpreted as the number of sets $R_i$ that contain $\boldsymbol{h}$:

$$\pi_{\mathcal{S}}(\boldsymbol{h}) = \frac{1}{m} |\{i : \boldsymbol{h} \in R_i\}|.$$

Notice that $R_i$ is bounded between two parallel affine hyperplanes :

$$A_i^+ = \{\boldsymbol{h} \in \mathbb{R}^n : (\boldsymbol{x}_i, 1) \cdot \boldsymbol{h} - y_i = p_i\}, \quad A_i^- = \{\boldsymbol{h} \in \mathbb{R}^n : (\boldsymbol{x}_i, 1) \cdot \boldsymbol{h} - y_i = -p_i\}.$$

A collection of affine hyperplanes is called *hyperplane arrangement*. For our purposes, given a sequence $\mathcal{S}$, we denote the induced hyperplane arrangement by

$$\mathcal{A}(\mathcal{S}) = \bigcup_{\substack{s \in \{\pm 1\} \\ 1 \leq i \leq m}} \{A_i^s\}.$$

A *region* of hyperplane arrangement $\mathcal{A}$ is a connected component in $\mathbb{R}^n \setminus \cup_{H \in \mathcal{A}} H$. We denote the set of regions of $\mathcal{A}$ by $\mathcal{R}(\mathcal{A})$, and the cardinality of $\mathcal{R}(\mathcal{A})$ by $r(\mathcal{A})$.

Observe that if $\boldsymbol{h}_1, \boldsymbol{h}_2$ are in the same region in $\mathcal{R}(\mathcal{A}(\mathcal{S}))$ it follows that they are mapped by $\mathcal{M}_{\mathcal{S}}^{\mathcal{H}}$ to the same vector $\boldsymbol{v}$. Therefore, $r(\mathcal{A})$ is essentially $|\mathcal{G}_{\mathcal{H}}(\mathcal{S})|$. Next, we bound $r(\mathcal{A})$:

**Lemma 5.** *Given hyperplane arrangement $\mathcal{A}$ that consists of $m$ pairs of parallel hyperplanes in $\mathbb{R}^n$, the maximal number of regions that can be formed is bounded by:*

$$r(\mathcal{A}) \leq \sum_{i=0}^{n} 2^i \binom{m}{i}.$$

*Proof.* We begin the proof by stating a known result in hyperplane arrangements:[3]

**Claim 1.** *Given an arrangement $\mathcal{A}$ and a hyperplane $H \notin \mathcal{A}$, define $\mathcal{A}^H = \{K \cap H : K \in \mathcal{A}\}$. It holds that:*
$$r(\mathcal{A} \cup \{H\}) = r(\mathcal{A}) + r(\mathcal{A}^H).$$

The proof of this claim appears in [14]. In order to bound the number of regions, we look at the worst case scenario, where the hyperplanes are in *generic* position, i.e. slightly moving a pair of parallel hyperplanes will not change the number of regions.

Next, we use inductive arguments - notice that $\mathcal{A}$ has two parameters, $n$ and $m$, thus we ought to have induction with two integer values. Denote:

$$T(m, n) = \text{The maximal number of regions induced by } m \text{ pairs of hyperplanes in } \mathbb{R}^n.$$

We seek to show that

$$T(m, n) = \sum_{i=0}^{n} 2^i \binom{m}{i}.$$

- *Base Cases*
    1. For $m \in \mathbb{N}_+$ it holds that $T(m, 1) = 2m + 1$, since each hyperplane is essentially a dot, and $2m$ dots cut $\mathbb{R}$ into $2m - 1$ bounded segments and two unbounded ones.
    2. For $m \in \mathbb{N}_+$ and $n = 2$, we need to show that $T(m, 2) = \sum_{i=0}^{2} 2^i \binom{m}{i} = 1 + 2m^2$. The base case for $m = 1$ is identical to the previous one, and assuming it holds for $m - 1$ parallel lines, each of the lines $\{A_m^-, A_m^+\}$ can intersect all other $2(m - 1)$ lines, thus
    $$T(m, 2) = 1 + 2 \cdot (m - 1)^2 + 2 \cdot (1 + 2m - 2) = 1 + 2m^2.$$

3. For $n \in \mathbb{N}_+$, $T(1, n) = \sum_{i=0}^{n} 2^i \binom{1}{i} = 3$, as any two parallel hyperplanes cut $\mathbb{R}^n$ into 3 sub-spaces.

- *Inductive step:* In case that the assumption is true for $T(m-1, n)$ and $T(m-1, n-1)$, we need to show that it also holds for $T(m, n)$. Denote by $\mathcal{A}$ an arrangement with $m-1$ pairs of hyperplanes in $\mathbb{R}^n$, and consider two additional parallel hyperplanes $A_m^+, A_m^-$. According to the claim above it follows that:

$$r\left(\mathcal{A} \cup \{A_m^+\}\right) = r(\mathcal{A}) + r\left(\mathcal{A}^{A_m^+}\right).$$

By invoking the claim one more time we get:

$$r\left(\mathcal{A} \cup \{A_m^+, A_m^-\}\right) = r\left(\mathcal{A} \cup \{A_m^+\}\right) + r\left(\left(\mathcal{A} \cup \{A_m^+\}\right)^{A_m^-}\right).$$

Observe that

$$\left(\mathcal{A} \cup \{A_m^+\}\right)^{A_m^-} = \left\{K \cap A_m^- : K \in \mathcal{A} \cup \{A_m^+\}\right\}$$

$$\overset{\substack{A_m^+, A_m^- \text{ are} \\ \text{parallel}}}{=} \left\{K \cap A_m^- : K \in \mathcal{A}\right\} = \mathcal{A}^{A_m^-}.$$

So altogether we know that

$$r\left(\mathcal{A} \cup \{A_m^+, A_m^-\}\right) = r(\mathcal{A}) + r\left(\mathcal{A}^{A_m^+}\right) + r\left(\mathcal{A}^{A_m^-}\right) \overset{\text{symmetry}}{=} r(\mathcal{A}) + 2r\left(\mathcal{A}^{A_m^+}\right).$$

Since $\mathcal{A}$ is composed of $(m-1)$ pairs of hyperplanes in $\mathbb{R}^n$, we have $r(\mathcal{A}) = T(m-1, n)$. In addition, since $A_m^+$ is a hyperplane in $\mathbb{R}^n$, it is isomorphic to $\mathbb{R}^{n-1}$, and every intersection of $A_m^+$ with $K \in \mathcal{A}$ is a hyperplane in $\mathbb{R}^{n-1}$. Hence, $r\left(\mathcal{A}^{A_m^+}\right) = T(m-1, n-1)$.

Finally:

$$
\begin{aligned}
T(m, n) &= r\left(\mathcal{A} \cup \{A_m^+, A_m^-\}\right) \\
&= r(\mathcal{A}) + 2r\left(\mathcal{A}^{A_m^+}\right) \\
&= T(m-1, n) + 2 \cdot T(m-1, n-1) \\
&\overset{\substack{\text{Inductive} \\ \text{assumption}}}{=} \sum_{i=0}^{n} 2^i \binom{m-1}{i} + 2\sum_{i=0}^{n-1} 2^i \binom{m-1}{i} \\
&= \sum_{i=0}^{n} 2^i \binom{m-1}{i} + \sum_{i=0}^{n} 2^i \binom{m-1}{i-1} \\
&\overset{\substack{\text{Pascal's} \\ \text{triangle}}}{=} \sum_{i=0}^{n} 2^i \binom{m}{i}.
\end{aligned}
$$

$\square$

**Proof of Theorem 1:** Follows directly from Lemma 5, since $\left|\mathcal{M}_{\mathcal{S}}^{\mathcal{H}}\right| = r(\mathcal{A})$.

**Another proof of Lemma 5:** The proof uses basic results in hyperplane arrangements, thus we start with a few definitions. Given an arrangement $\mathcal{A}$, the partially ordered set (poset) $\mathcal{I}(\mathcal{A})$ is defined as the set of all possible intersection of elements in $\mathcal{A}$, where $\mathbb{R}^n \in \mathcal{I}(\mathcal{A})$ as the empty intersection. The binary relation $\leq$ is inverse inclusion, where $A \subseteq B$ if and only if $A \geq B$, and the minimal element in $\mathcal{I}(\mathcal{A})$ is $\mathbb{R}^n$, denoted $\hat{0}$. The *mobius function* $\mu$ on the poset $\mathcal{I}(\mathcal{A})$ is defined recursively by the two following properties:

- $\mu(x, x) = 1$ for all $x \in \mathcal{I}(\mathcal{A})$.

- $\sum_{x \leq z \leq y} \mu(x, z) = 0$ for all $x < y$ in $\mathcal{I}(\mathcal{A})$.

In order to bound the number of regions, we look at the worst case scenario, where the elements of $\mathcal{A}(\mathcal{S})$ are in *generic* position, .i.e slightly moving a hyperplane will not change the number of regions. Next, we claim that $\mu(\hat{0}, y) = (-1)^{n-dim(y)}$. This is a standard result in enumerative combinatorics, but appears here for completeness. Denote $rk(y) = n - dim(y)$. Clearly, if $rk(y) = 0$, we have $y = \hat{0} = \mathbb{R}^n$ thus the claim holds by definition of $\mu$. For $y$ such that $rk(y) = k > 0$, $y$ is the intersection of $k$ hyperplanes in $\mathcal{A}(\mathcal{S})$, and since these hyperplanes in generic position it follows that the number of $z \in \mathcal{I}(\mathcal{A}(\mathcal{S}))$ such that $rk(z) = i$ and $z \leq y$ is $\binom{k}{i}$. Thus

$$\sum_{\hat{0} \leq z \leq y} \mu(\hat{0}, z) = \sum_{i=0}^{k} (-1)^i \cdot \binom{k}{i} = 0,$$

hence proving the claim. The *characteristic polynomial* associated with $\mathcal{A}$ is defined as:

$$\chi(\mathcal{A}, q) = \sum_{x \in \mathcal{I}(\mathcal{A})} \mu(\hat{0}, x) \cdot q^{dim(x)}.$$

Observe that the number of elements $x$ in $\mathcal{I}(\mathcal{A}(\mathcal{S}))$ such that $rk(x) = i$ is $2^i \cdot \binom{m}{i}$, since we can first choose $i$ points out of $m$ and afterwards decide one of the two hyperplanes associated with point $j$ we choose (whether we take $A_j^+$ or $A_j^-$). As a result, the characteristic polynomial of $\mathcal{I}(\mathcal{A}(\mathcal{S}))$ is:

$$\chi(\mathcal{I}(\mathcal{A}(\mathcal{S})), q) = \sum_{i=0}^{n} 2^i \binom{m}{i} \cdot (-1)^{n-i} \cdot q^{n-i} = \sum_{i=0}^{n} 2^i \cdot (-q)^{n-i} \binom{m}{i}.$$

Finally, due to Zaslavsky's theorem [19] we know that $r(\mathcal{A}) = \chi(\mathcal{A}, -1)$, therefore:

$$r(\mathcal{A}(\mathcal{S})) = \chi(\mathcal{I}(\mathcal{A}(\mathcal{S})), -1) = \sum_{i=0}^{n} 2^i \binom{m}{i}.$$

$\square$

**Proof of Lemma 2**   Recall that:

$$\forall h \in \mathcal{H}, \forall z \in \mathcal{Z} : g_h(z) = \begin{cases} 1 & |h(x) - y| \geq p \\ 0 & |h(x) - y| < p \end{cases}.$$

Fix a sample set $\mathcal{S}$ of size $m$, and denote

$$\mathcal{G}_{\mathcal{H}}(\mathcal{S}) = \{(g_h(c_1), g_h(c_2), \ldots, g_h(c_m)) : c_i = (x_i, y_i, p_i), g_h \in \mathcal{G}_{\mathcal{H}}\}.$$

If $\mathcal{G}_{\mathcal{H}}(\mathcal{S})$ is shattered, it follows that

$$2^m \leq r(\mathcal{A}(\mathcal{S})) = \sum_{i=0}^{n} 2^i \binom{m}{i}.^4$$

By the Sauer - Shelah lemma [12] we know that $\sum_{i=1}^{n} \binom{m}{i} \leq \left(\frac{em}{n}\right)^n$. Hence:

$$2^m \leq \sum_{i=0}^{n} 2^i \binom{m}{i} \leq \left(\frac{2em}{n}\right)^n. \tag{3}$$

Conversely, if Equation (3) does not hold for some $m$, then $m$ is an upper bound of VCdim$(\mathcal{G}_{\mathcal{H}})$. When plugging $m = 2n \log(n)$ on the negation of Equation (3) we get

$$2^{2n \log(n)} > \left(\frac{4en \log(n)}{n}\right)^n \Rightarrow \left(2^{\log(n^2)}\right)^n > (4e \log(n))^n \Rightarrow n^2 > 4e \log(n),$$

and the latter holds for all $n \geq 6$. In addition, it can be verified that $2^{20} > \sum_{i=1}^{5} 2^i \binom{20}{i}$, thus VCdim$(\mathcal{G}_{\mathcal{H}}) \leq 20$ if $n \leq 5$.

$\square$

**Proposition 2.** *The Partial Vector Feasibility problem is solvable in polynomial time in $m$ and $n$.*

*Proof.* Consider the following LP:

$$\max_{\boldsymbol{h},\epsilon} \epsilon$$

subject to

$$
\begin{aligned}
\boldsymbol{h} \cdot (\boldsymbol{x}_i, 1) - y_i &\leq p_i - \epsilon && \text{for } u_i = 1 \\
\boldsymbol{h} \cdot (\boldsymbol{x}_i, 1) - y_i &\geq -p_i + \epsilon && \text{for } u_i = 1 \\
\boldsymbol{h} \cdot (\boldsymbol{x}_i, 1) - y_i &\geq p_i + \epsilon && \text{for } u_i = a \text{ (above)} \\
\boldsymbol{h} \cdot (\boldsymbol{x}_i, 1) - y_i &\leq -p_i - \epsilon && \text{for } u_i = b \text{ (below)} \\
\epsilon &\leq B && (B \text{ is a large constant})
\end{aligned}
$$
(P1)

The vector $\boldsymbol{u}$ is feasible if $\epsilon$ is greater than zero. If so, return $\boldsymbol{h}$, and otherwise return $\phi$ $\qquad\square$

**Proof of Theorem 2:** Observe that EPM performs recursive partitioning of $\mathbb{R}^n$: in each iteration of the for loop in Line 3, it partitions the subspace that corresponds to the partial vector $\boldsymbol{v}$ according to the affine hyperplanes induced by the $i$'th point in $\mathcal{S}$. Thus, in Line 9 it considers the payoff of all the regions in $\mathcal{R}\left(\mathcal{A}(\mathcal{S})\right)$, and returns a feasible vector with the highest payoff. It is easy to choose a best response once we have $\boldsymbol{v}^*$: we just run PVF$(\mathcal{S}, \boldsymbol{v}^*)$ and pick the vector $\boldsymbol{h}$ achieving the optimal solution.

The for loop in Line 3 iterates $m$ times. For each $i$, the algorithm iterates through all elements in $\mathcal{R}_{i-1}$, and for each element solves three instances of PVF. Thus, if we show that for all indices $i$ $\mathcal{R}_i$ is poly$(m)$, we will deduce that the time complexity of EPM is poly$(m)$. Indeed, by Lemma 5 we know that $|\mathcal{R}_m| \leq r\left(\mathcal{A}(\mathcal{S})\right)$, so it is sufficient to show that

$$|\mathcal{R}_0| \leq |\mathcal{R}_1| \cdots \leq |\mathcal{R}_m|.$$

We use inductive arguments: the base case is true because $|\mathcal{R}_0| = 1$ and $|\mathcal{R}_1| = 3$, since $(\alpha, 0, \ldots 0)$ is feasible for all $\alpha \in \{2, 1, -2\}$. Assume that the claim is correct for $i - 1$, and fix $\boldsymbol{v} \in \mathcal{R}_{i-1}$. Since $\boldsymbol{v}$ defines a subspace in $\mathbb{R}^n$, which is the union of the sub-spaces that correspond to the vectors $(\boldsymbol{v}_{-i}, 1), (\boldsymbol{v}_{-i}, a), (\boldsymbol{v}_{-i}, b)$ along with $A_i^+, A_i^-$, at least one of these vectors is feasible. As a result, $\boldsymbol{v}$ contributes at least one element to $\mathcal{R}_i$. This concludes the proof of Theorem 2. $\qquad\square$

**Proof of Lemma 3:** If the opponent is using a linear strategy, by definition of $A_i^+, A_i^-$, every triplet in $(x_i, y_i, z_i) \in \mathcal{S}$ is either in $A_i^+$ or $A_i^-$. Observe that

**Claim 2.** *If $\boldsymbol{h}^* \in \arg\max_{\boldsymbol{h}' \in \mathbb{R}^n} \pi_{\mathcal{S}}\left(\boldsymbol{h}'\right)$, there exists a region $F \in \mathcal{R}\left(\mathcal{A}(\mathcal{S})\right)$ such $\boldsymbol{h}^* \in F$ and $\bar{\boldsymbol{h}}$ is on the exterior of $F$.*

This is proved by contradiction: suppose the region $\boldsymbol{h}^*$ lies in does not have $\bar{\boldsymbol{h}}$ on its exterior. By definition of $A_i^+, A_i^-$, every triplet in $(x_i, y_i, z_i) \in \mathcal{S}$ is either in $A_i^+$ or $A_i^-$, thus there exist a region $F'$ with $\bar{\boldsymbol{h}}$ on its exterior. Fix a point $\boldsymbol{h}'$ on the line between $\bar{\boldsymbol{h}}$ and $\boldsymbol{h}^*$, such that $\boldsymbol{h}' \in F'$.

Since $\boldsymbol{h}'$ and $\boldsymbol{h}^*$ are not in the same region, when traveling on the straight line from $\boldsymbol{h}'$ to $\boldsymbol{h}^*$ we must cross at least one hyperplane, denoted $A_i^\sigma$. Since $A_i^{-\sigma}$ passes in $\bar{\boldsymbol{h}}$, it follows that $\boldsymbol{h}'$ is inside $R_i$, and that $\boldsymbol{h}^*$ is outside $R_i$. Hence, for every hyperplane we cross on the traversal from $\boldsymbol{h}'$ to $\boldsymbol{h}^*$ we necessarily lose the $i$'th point since we exit $R_i$, thus achieving the desired contradiction.

Next, denote $\hat{\boldsymbol{h}} = \lambda \cdot \boldsymbol{h}^* + (1 - \lambda) \cdot \bar{\boldsymbol{h}}$, for $\lambda \in (0, 1)$. Since $h'$ and . Since $\hat{\boldsymbol{h}}, \boldsymbol{h}^*$ are in the same region, $\pi_{\mathcal{D}}(\hat{\boldsymbol{h}}) = \pi_{\mathcal{D}}(\boldsymbol{h}^*)$. In addition:

$$
\begin{aligned}
\text{MEANERROR}(\hat{\boldsymbol{h}}, \mathcal{S}) &= \frac{1}{m} \sum_{i=1}^{m} \left\| \hat{\boldsymbol{h}} \cdot (\boldsymbol{x}_i, 1) - y_i \right\| \\
&= \frac{1}{m} \sum_{i=1}^{m} \left\| \lambda \left(\boldsymbol{h}^* \cdot (\boldsymbol{x}_i, 1) - y_i\right) + (1 - \lambda) \left(\boldsymbol{h}^* \cdot (\boldsymbol{x}_i, 1) - y_i\right) \right\| \\
&\leq \lambda \cdot \text{MEANERROR}(\boldsymbol{h}^*, \mathcal{S}) + (1 - \lambda) \cdot \text{MEANERROR}(\bar{\boldsymbol{h}}, \mathcal{S}).
\end{aligned}
$$

Thus MEANERROR$(\hat{\boldsymbol{h}}, \mathcal{S})$ can be arbitrarily close to MEANERROR$(\bar{\boldsymbol{h}}, \mathcal{S})$, by setting $\lambda$ properly. Due to uniform converges on $\mathcal{H}$, the same principle holds w.r.t. the distribution as well.

**Proof of Lemma 4:**   Let

$$\text{LINEARRESPONSE} = \left\{ (\mathcal{S}, k) : \exists \boldsymbol{h} \in \mathbb{R}^n, \pi_{\mathcal{S}}(\boldsymbol{h}) = \frac{k}{|\mathcal{S}|} \right\}.$$

We show polynomial time reduction from the Independent Set problem, which is known to be NP-complete (see e.g. [5]), and is defined by

$$\text{INDSET} = \{ (V, E, k) : G = (V, E) \text{ has an indepenent set of size } k \}.$$

Given $(V, E, k)$, we construct $(\mathcal{S}, k')$ such that $k' = (|V| + 1) \cdot (|E| + 1) + k$ and $\mathcal{S}$ induces the indicator functions that correspond to

$$
\begin{array}{llll}
\text{(Type 1)} & \forall (v_i, v_j) \in E & |h_i + h_j + h_0 + 1| < 1.5 \\
\text{(Type 2)} & \forall v_i \in V & |h_i + h_0 - 1| < 0.5 \\
\text{(Type 3)} & & |h_0 + 0.5\epsilon| < 0.5\epsilon
\end{array}
$$

for some small positive rational constant $\epsilon$. We duplicate $|V| + 1$ times every point of Types 1 and 3. Indeed, the empirical payoff can be constructed by this set of constraints, and the reduction is polynomial.

Let $(V, E, k) \in \text{INDSET}$, and denote an arbitrary independent set by $I \subseteq V$. Define $\boldsymbol{h} = (h_0, h_1, \dots h_{|V|})$ such that:

$$
h_i = \begin{cases} -\frac{\epsilon}{2} & i = 0 \\ 1 & v_i \in I \\ -1 & v_i \notin I \end{cases}.
$$

$\boldsymbol{h}$ gains Type 2 point if and only if it corresponds to a vertex in $I$. In addition, all Type 1 points are gained by $\boldsymbol{h}$ since at most one of $\{v_i, v_j\}$ is in $I$ if $(v_i, v_j) \in E$. Clearly, all Type 3 are gained as well. As a result $\pi_{\mathcal{S}}(\boldsymbol{h}) = (|V| + 1) \cdot (|E| + 1) + k = k'$, thus $(\mathcal{S}, k') \in \text{LINEARRESPONSE}$.

Conversely, assume $(\mathcal{S}, k') \in \text{LINEARRESPONSE}$. By definition of LINEARRESPONSE there exists $\boldsymbol{h}$ which grabs all points of Types 1 and 3, since each Type $\{1, 3\}$ point has more than $|V|$ copies, whereas $\boldsymbol{h}$ loses less than $|V|$ points. We consider the set $I \subseteq V$ containing all nodes that correspond to Type 2 points gained by $\boldsymbol{h}$. Hence $|I| = k$, and we verify that $I$ is an independent set: suppose that $I$ contains $(v_i, v_j) \in E$. If so, the Type 2 points corresponding to $v_i, v_j$ are gained by $\boldsymbol{h}$, thus

$$
\begin{cases} |h_i + h_0 - 1| < 0.5 \\ |h_j + h_0 - 1| < 0.5 \end{cases} \Rightarrow 1 + \epsilon < h_i + h_j < 3 + \epsilon. \tag{4}
$$

On the other hand, $\boldsymbol{h}$ gains all Type 1 points, and in particular the point associated with the edge $(v_i, v_j)$, hence

$$
\left| h_i + h_j - \frac{\epsilon}{2} + 1 \right| < 1.5 \Rightarrow -2.5 + \frac{\epsilon}{2} < h_i + h_j < 0.5 + \epsilon. \tag{5}
$$

Since Equations (4) and (5) cannot be satisfied at the same time, the assumption is false and $I$ is an independent set. Thus, $(V, E, k) \in \text{INDSET}$.

$\square$

# 7   Mixed integer linear programming formulation

We slightly change the payoff function, by introducing a prediction tolerance in the following sense: a point $(x_i, y_i)$ will be associated with the agent if and only if $|h(x_i) - y_i| \leq p_i - \epsilon$, for some small and fixed $\epsilon$. As a result, the empirical payoff of the agent is now defined as:

$$
\pi_{\mathcal{S}} \left( \bar{h}, \boldsymbol{h} \right) = \frac{1}{m} \sum_{i=1}^{m} \mathbb{1}_{|\boldsymbol{h} \cdot (\boldsymbol{x}_i, 1) - y_i| \leq p_i - \epsilon}. \tag{6}
$$

For each summand of the sum in Equation (6), we define a binary variable $z_i$ as follows:

$$
z_i = \begin{cases} 0, & \text{if } |\boldsymbol{h} \cdot (\boldsymbol{x}_i, 1) - y_i| \leq p_i - \epsilon \\ 1, & \text{otherwise} \end{cases}.
$$

Next, we formulate the problem of maximizing the expression in Equation (6) as -

$$\min_{\boldsymbol{h}, z_1, \ldots, z_m} \quad \sum_{i=1}^{m} z_i$$

subject to
$$y_i - \boldsymbol{h} \cdot (\boldsymbol{x}_i, 1) - B \cdot z_i \le p_i - \epsilon, \qquad i = 1, \ldots, m$$
$$-y_i + \boldsymbol{h} \cdot (\boldsymbol{x}_i, 1) - B \cdot z_i \le p_i - \epsilon, \qquad i = 1, \ldots, m$$
$$z_i \in \{0, 1\}, \qquad i = 1, \ldots, m$$

where B is chosen to be large enough so that the constraints for the $i$'th point will always hold if $z_i = 1$, namely $|y_i - \boldsymbol{h} \cdot (\boldsymbol{x}_i, 1)| \le p_i - \epsilon + B$ for all $\boldsymbol{h} \in \mathbb{R}^n, i \in \{1, \ldots, m\}$. Note that it has $3m$ constraints and $m + n$ variables.