[Reviews · NeurIPS 2017]

Reviewer 1



Setting: There are two regression learners, and "opponent" and "agent". The opponent has some fixed but unknown hypothesis. The agent observes samples consisting of data pairs x,y along with the absolute loss of the opponent's strategy. The agent will produce a hypothesis. On a given data point, the agent "wins" if that hypothesis has smaller absolute loss than the opponent's. The goal is to win with highest probability on the underlying distribution. Results: Proposes the problem. Reduces the problem to PAC learning, working out the linear regression case. Some experiments for this case. My opinion: I think this is an interesting paper raising an interesting and possibly important question. The results make at least a good first step in answering the question. I slightly worry about overlap with the dueling algorithms literature, but think the learning problem proposed here is in the end quite different. Comments for authors: I find several things interesting about your approach and questions raised by the paper. 1. Extensions to other goals than minimum absolute error could be interesting. For example, if h(x) is a ranking of alternatives, then we could say the agent "wins" if her ranking has some property compared to the opponents'. I think this would touch on the dueling algorithms literature too. One could also not have a "winner take all" reward but instead some continuous reward between 0 and 1 as a function of the two algorithms' predictions. 2. I think the approach of [treating the agent's learning problem as binary classification] is very interesting. On one hand, it makes perfect sense to pair with 0-1 loss. On the other hand, it seems to throw away some useful information. Intriguing. 3. The game theory/equilibrium extensions sounds difficult because of the machine-learning setting formulation where the underlying distribution is initially unknown and accessible via samples. Usually to formalize equilibrium one needs the agents to form consistent Bayesian beliefs about the unknown and best-respond to those beliefs. Would be very interesting if this modeling challenge could be overcome.

Reviewer 2



The paper introduces a 'duel' version of linear regression, where the learner tries to out-predict the opponent on the most data points, in a PAC sense, as opposed to simply minimizing expected loss overall. The authors give an algorithm to compute the best linear response against any competitor, show that the problem of finding such a best response can be NP-hard, and demonstrate their algorithm empirically. Overall, I think this paper is interesting, and the results thorough. I appreciated the attention to exposition. I was confused about two points, which I would like the authors to address in the rebuttal: 1. It is assumed that the best-responder knows the discrepancies but not the hypothesis \bar h, nor the signed discrepancies. Why is this reasonable? In the real-estate example, the actual predictions of the opponent are known, and this seems to be a more typical case. Perhaps the hardness could be circumvented in this more powerful model (though I somewhat doubt it). 2. I did not find the authors' justification for why one would not worry about "embarrassing predictions" to be very satisfying. The authors say that predicting poorly on a handful of data points is not a concern in "most" practical scenarios, but only a vague example about "demographics" is given, leaving one to conjure up countless scenarios where this is indeed a problem (consider the backlash toward xbox for failing to recognize gestures from minority users, ditto for speech recognition, etc etc). In spite of this seemingly "straw man" argument, however, it seems that the algorithm can be made to avoid embarrassing predictions, at least relative to a linear opponent (Lemma 3). Hence, I would view that result as satisfying, and the lack of such a result for a non-linear opponent concerning. line 119: Make this a full sentence line 123: In bullet 3 of the model, it is said that \bar h is not known to the best-responder, but of course something must be known, and later in that paragraph it is mentioned that p = |\bar h - y| is known. It would be much clearer to state up front what IS assumed to be known, before assuring the reader what is not assumed to be known.

Reviewer 3



STRENGTHS: On some "abstract" level, this paper pursues an interesting direction: - I fully agree that "prediction is not done in isolation" (l24), and though this is a vague observation, it is worth pursuing this beyond established work (although I'm not an expert with a full overview over this field). - Making explicit the tradeoff between good-often and not-bad-always in deciding on a loss function is interesting. Overall, the writing and mathematical quality of the paper is good to very good, with some uncertainty regarding proofs though: - Well written and structured. - The paper is very strong in that the algorithmic and mathemtaical results are well connected (combine the VC theorems and lemmas to reach corollary 1 which shows sample bounds for their EPM algo) and cover the usual questions (consistency, runtime). - In terms of correctness of the mathematical results, it seems that the authors know what they are doing. I only checked one proof, that of Theorem 1 (via the second proof of Lemma 5), and it seems correct although I couldn't follow all steps. WEAKNESSES: For me the main issue with this paper is that the relevance of the *specific* problem that they study -- maximizing the "best response" payoff (l127) on test data -- remains unclear. I don't see a substantial motivation in terms of a link to settings (real or theoretical) that are relevant: - In which real scenarios is the objective given by the adverserial prediction accuracy they propose, in contrast to classical prediction accuracy? - In l32-45 they pretend to give a real example but for me this is too vague. I do see that in some scenarios the loss/objective they consider (high accuracy on majority) kind of makes sense. But I imagine that such losses already have been studied, without necessarily referring to "strategic" settings. In particular, how is this related to robust statistics, Huber loss, precision, recall, etc.? - In l50 they claim that "pershaps even in most [...] practical scenarios" predicting accurate on the majority is most important. I contradict: in many areas with safety issues such as robotics and self-driving cars (generally: control), the models are allowed to have small errors, but by no means may have large errors (imagine a self-driving car to significantly overestimate the distance to the next car in 1% of the situations). Related to this, in my view they fall short of what they claim as their contribution in the introduction and in l79-87: - Generally, this seems like only a very first step towards real strategic settings: in light of what they claim ("strategic predictions", l28), their setting is only partially strategic/game theoretic as the opponent doesn't behave strategically (i.e., take into account the other strategic player). - In particular, in the experiments, it doesn't come as a complete surprise that the opponent can be outperformed w.r.t. the multi-agent payoff proposed by the authors, because the opponent simply doesn't aim at maximizing it (e.g. in the experiments he maximizes classical SE and AE). - Related to this, in the experiments it would be interesting to see the comparison of the classical squared/absolute error on the test set as well (since this is what LSE claims to optimize). - I agree that "prediction is not done in isolation", but I don't see the "main" contribution of showing that the "task of prediction may have strategic aspects" yet. REMARKS: What's "true" payoff in Table 1? I would have expected to see the test set payoff in that column. Or is it the population (complete sample) empirical payoff? Have you looked into the work by Vapnik about teaching a learner with side information? This looks a bit similar as having your discrapency p alongside x,y.